# Development and Application of AR-Based Assessment System for Infant Airway Obstruction First Aid Training

**DOI:** 10.3390/children9111622

**Published:** 2022-10-26

**Authors:** Pao-Ju Chen, Wei-Kai Liou

**Affiliations:** 1Department of Nursing, College of Healthcare and Management, Asia Eastern University of Science and Technology, New Taipei City 220303, Taiwan; 2College of Management and Design, Ming-Chi University of Technology, New Taipei City 243303, Taiwan

**Keywords:** augmented reality, foreign body airway obstruction, first aid training, assessment system

## Abstract

Airway obstruction refers to suffocation caused by blockage of the airway due to a foreign body and is a common cause of accidental death in infants below one year of age. However, the current infant CPR manikins used for training in first aid for foreign body airway obstruction can only be applied to one single scenario. Furthermore, trainees’ first aid skills cannot be recorded and quantified with a digital system and, consequently, assessment of their skills is difficult to conduct. This study aims to overcome the technical limitations by developing an AR-based assessment system for training in first aid for infant airway obstruction. With this assessment system, trainees can learn first aid more efficiently and correctly and conduct a quantitative assessment of their skills digitally. For instructors, the time required for assessment, potential human error, and the cost of training can also be reduced. The system can be a practical learning tool that helps trainees assess emergencies and integrate their knowledge and skills.

## 1. Introduction

First aid is one of the important professional skills required of nurses. Competence in performing first aid allows one to provide rescue immediately during an emergency, which can avoid missing the urgent time that is crucial to an injured person’s survival and reduce the risk of unfortunate incidents. Airway obstruction refers to suffocation caused by blockage of the airway due to a foreign body and is a common cause of accidental death in infants [1]. Typically, first aid is taught and demonstrated by instructors in person. Due to the crowded space in the classroom, not all students can see the demonstration by the instructors clearly, and the demonstration may also be interrupted by students.

Hence, students may still lack confidence in performing first aid and feel anxious even after they have completed the training. Most of the current first aid training for airway obstruction in infants under one year of age involves delivering back slaps and chest thrusts to a manikin representing a choking infant [1]. However, there are three limitations to this type of training. First, infant choking manikins can only be used to practice delivering back slaps and chest thrusts. Second, the training can only be conducted on site. Third, the trainees’ first aid skills cannot be recorded and quantified digitally, making it difficult for instructors to offer feedback. Since infants are unable to express themselves verbally, first aiders need to provide immediate assistance according to the infant’s symptoms and reactions when airway obstruction occurs. Therefore, it is important to familiarize trainees with the various symptoms of infant airway obstruction during first aid training [2].

With the rapid development of technology in recent years, flipped classrooms have been widely implemented. In nursing education, the traditional teaching approach that emphasizes lecture-based instruction and demonstration has also evolved into a self-learning approach that helps students integrate knowledge, emotions, and professional skills to keep up with clinical practice. Moreover, due to the coronavirus pandemic, schools have been forced to move classes online to reduce the risk of transmission, and the need for resources that support distance learning has therefore increased. In this case, first aid training can be integrated with digital media, including mobile devices, software and hardware, and applications, so that more stimuli, such as sounds, images, and materials, are provided to learners [3].

Augmented reality (AR) combines computer-generated images, objects, information, or scenarios with the real-world environment and provides interactive experiences to users. AR incorporates three features: a combination of real and virtual worlds, real-time interaction, and a requirement for a 3D space [2]. Combining scenario-based learning with AR technology has been one of the teaching approaches implemented on site [4,5]. Through an AR-based assessment system that allows digital interaction, learners can learn first aid more efficiently and correctly while at the same time conducting a quantitative assessment of their skills. The system can not only be applied to classroom and on-the-job training in medical institutions but could also be used as a tool for distance learning during the pandemic. The advantages of AR in nursing education include a safe environment provided to learners; the convenience of delivering lectures; accurate information; and accessibility to instruction at any time and from anywhere [6]. Numerous studies on the adoption of AR in teaching indicate that the integration of AR technology with learning is meaningful since students’ academic achievements and learning motivation have increased [7]. Moreover, students’ anxiety about learning has decreased and their learning satisfaction has improved [8]. Hence, it is clear that AR is an effective learning tool for first aid training [9].

Because of the unique anatomy and physiology of infants and toddler children, airway obstruction by foreign bodies is one of the leading causes of accidental death [10]. Besides complete foreign body airway obstruction that is immediately life-threatening, partial airway obstructions can impede gas exchange and lead to dyspnea and pneumonia [11]. Thus, early recognition, emergency care, and management are essential. The objectives of this study are to develop an AR-based assessment system for training in first aid for infant airway obstruction and an interactive manikin. This system, combined with the manikin, can guide learners through first aid training step-by-step and offer immediate feedback to ensure that their procedures are completely correct, and then record and quantify their performance and provide immediate evaluation. Furthermore, this study also attempted to understand students’ learning experiences toward self-learning using the AR skill education system.

## 2. Materials and Methods

### 2.1. Design and Participants

This study employs a qualitative research design and the research method of focus group interview, selecting a sample of 82 students from the Department of Nursing at a university in New Taipei City, Taiwan, by purposive sampling. The participants’ inclusion criteria are: a. those who were willing to participate in this study and signed the informed consent form; b. those who have normal cognitive function, no visual or hearing impairment, and can express themselves fluently through spoken language; c. the participants who were able to participate in the entire infant first aid training course and the focus group interviews.

### 2.2. Infant Airway Obstruction First Aid Basic Skills

First aid skills for infant airway obstruction were planned as part of the infant first aid training module of the pediatric nursing practical course. The instructor teaches students first aid skills for airway obstruction in infants based on the guidelines for providing emergency care for infants from the 2020 American Heart Association guidelines [12]. A child with a presumed airway obstruction that is still able to maintain some degree of ventilation should be allowed to clear the airway by coughing. If the child cannot cough, vocalize, or breathe, emergent steps are necessary to clear the airway. For infants under one year of age, alternating sequences of five back blows and five chest thrusts are performed until the object clears or the infant becomes unresponsive. Abdominal thrusts should not be performed on infants as their livers are more prone to injury.

### 2.3. Data Collection Procedure

The course instructor first demonstrated the first aid procedure and the key points of back blows and chest thrusts, emphasizing that the infant’s head and neck must be sufficiently fixated with one hand during the first aid procedure to avoid intracranial hemorrhage due to shaking. Students were allowed to practice with traditional infant choking manikins. Afterwards, students would experience the “AR Infant Airway Obstruction First Aid Assessment System” and repeat the exercise using the interactive model developed in this study. At the end of the practical sessions, the research method of focus group was used to understand individual attitudes, perspectives, or beliefs about a topic, thus allowing researchers to gain insight into interesting situations and to clarify the meaning behind certain behaviors [13]. The focus group interviews were held from the 12th to the 14th of April in 2022 and consisting of six 50-min sessions with approximately 13 to 14 participants. The interviews took place in a quiet and comfortable discussion room. In order to protect the rights of the students, the informed consent form states that students’ participation in AR learning and interviews is not related to their academic performance.

During the interviews, the students were asked about their perceptions of utilizing the AR system to learn first aid skills and reflect on its impact on learning. The researchers confirmed the content and the perceptions provided by the participants and ensured that the participants had sufficient time to fully answer each question, thus ensuring the credibility of the study. In addition to discussions on the content, the interview records also included the participants’ nonverbal behavior or emotions. The interview outline was first developed by the researchers through a literature review; subsequently, the interview questions were finalized based on their individual teaching experience and the consensus of the research team. The course instructor was not the moderator of the focus group interview sessions. The moderator of the interview sessions was another member of the research team who was not the instructor of the first aid course.

The guiding questions of the interview are as follows:Please share your feelings about the learning experience of first aid skills through the AR system.What is the difference between practicing first aid skills and traditional skills in an AR learning environment?How did the AR system help or benefit you in learning first aid skills?Did you encounter any difficulties in learning first aid skills with the AR system?In the face of the trend of integrating digital technology into the curriculum, do you have any suggestions for the usage of AR in teaching?

### 2.4. Data Analysis

In this study, content analysis was used to analyze the data, and the steps were as follows: first, the audio recordings of the focus sessions were translated verbatim without any changes or additions; afterwards, the researchers read the data repeatedly to search for the categories presented in the data and categorize the data; lastly, during the coding process, the codes of the same nature were connected to form categories, and the links between the categories were further unified to form the main theme [14,15]. Two experts trained in qualitative research were invited to conduct a peer examination, and five interviewees were asked to conduct a member check to confirm whether the data analysis results represented their real learning experiences. The NVivo 10 software program was used to code the data according to the themes and extract key data from each theme.

### 2.5. Development of AR System

The AR system simulates the scenarios of an infant choked by a foreign body, helping learners make correct judgments and remove the foreign body from the infant’s throat during an urgent time. Through the system’s information integration and interactive feedback mechanism, learners can integrate their knowledge and skills. To develop the AR system, Unity, a cross-platform game engine, and Vuforia AR are used to create an SDK according to the learning goals and content. With the Unity engine, AR applications can be created quickly on both the Android and iOS platforms. During a first aid training session, learners need to make the correct judgment and perform first aid according to the injured person’s symptoms. Through the system’s integration of real-world and virtual information and the interactive feedback mechanism, learners are provided with a practical learning tool to integrate their knowledge and skills. Its principles and processes are explained as follows.

#### 2.5.1. AR Scanning System

Vuforia AR and Unity create image targets and generate AR scenes. After the setting is done, the infant’s symptoms, reactions, and sounds can be observed by users. Vuforia creates AR with its object recognition and image tracking features. Learners use their tablet’s camera to scan the infant manikin. The scanned image is then transferred to the system database and a 3D image of the simulated scenario is generated and overlaid over the real-life objects, which is then displayed on the tablet’s screen (Figure 1).

#### 2.5.2. Pressure Sensor Detecting Back Slaps

When back slaps are delivered, the pressure sensor on the back of the infant manikin sends electronic signals to the system. If the back slaps are delivered to the right position and with the correct frequency, positive feedback will show on the screen and a light will show; if the back slaps are delivered incorrectly, the screen will indicate an error and no light will be shown (Figure 2).

#### 2.5.3. Pressure Sensor Detecting Chest Thrusts

When CPR is being performed, chest thrusts should be delivered at a rate of 100–120 times per minute and to a depth of 4–5 cm. When chest thrusts are delivered, the pressure sensor on the chest of the manikin sends electronic signals to the system. If the chest thrusts are delivered to the right position and with the correct frequency, positive feedback will show on the screen and the light will show; if the chest thrusts are delivered incorrectly, the screen will indicate an error and no light will be shown (Figure 3).

#### 2.5.4. Wireless Transmission System

Electronic signals are sent from the pressure sensors and transmitted to the handheld device through the wireless transmission system; the application then judges whether the first aid is performed correctly and displays the result on the screen of the device (Figure 4).

#### 2.5.5. System Operation Process

The first step is to scan the QR code with a mobile phone to download the first aid assessment system application and then click the screen to enter the start page. If a learner can correctly assess the infant’s airway blockage, they should then proceed to the interactive infant manikin to perform first aid. To perform first aid, they should pick up the manikin and turn it over to deliver five back slaps, and then turn it back to the front and deliver five chest thrusts. The sensors on the manikin detect a user’s movements, such as flipping, moving, and pressing, and will send electronic signals accordingly. The sensors can even detect the sequence, strength, and frequency of a user’s movements. If back slaps and chest thrusts are performed correctly, the light will be displayed every time. On the contrary, the light is not displayed if first aid is performed incorrectly. The database of the back-end system contains voice prompts for users. If the back slaps and chest thrusts are delivered correctly, positive feedback will show on the handheld device’s screen and there will be a voice prompt telling users to proceed to the next step. If the back slaps and chest thrusts are delivered incorrectly, the screen will indicate an error and there will be a voice prompt telling users to repeat the same step, which will continue until it is done correctly.

In general, learners need to know how to assess a clinical situation and provide correct first aid treatment in a real-world environment. If a learner cannot perform first aid correctly or during an urgent time, the AR assessment system will give corresponding feedback and sound a warning so that the learner can be corrected immediately. Through the system, learners are allowed to interact with images and sounds and can practice first aid repeatedly to deepen their impressions. The system provides a learning environment that is effective, interactive, and intuitive (Appendix A). System operation process video links: https://www.youtube.com/watch?v=pzZ_dQaEFi4, accessed on 28 April 2022.

### 2.6. Ethical Considerations

Written approval for this study was obtained from the university ethics committee (8 November 2021, Decision number: NCCU-REC-202108-E088). Participants were informed about the purpose and plan of the study, and written consent was obtained from all students. Additionally, the participants were made aware that the group meeting process would be recorded and that they had the right to withdraw from the study or halt the recording process during the interview. The content of the interview was processed anonymously.

## 3. Results

This AR system was applied to the child nursing curriculum in April 2022. Among the post-course reflections of 82 nursing students in the nursing department of a university in New Taipei City, Taiwan, most students provided positive feedback on this AR teaching system. The following categories are summarized from student reflections.

### 3.1. Inspiring Empathy

The AR system can stimulate users’ senses and provide an experience merging the virtual and real worlds, which helps stimulate ideas in learners. At the same time, learners feel more present through AR and therefore build more empathy.

Two students stated the following:

“The baby’s painful expression and the sound of struggling due to suffocation could be seen and heard from the phone at the same time, which shocked me and made me feel an urge to perform first aid immediately.”

“On the phone screen, the baby’s lips were turning blue in the beginning. But after I performed first aid correctly, I felt that I had saved the baby’s life, which is a unique experience.”

### 3.2. Improving Clinical Judgment and Decision-Making Skills

The AR system provides various simulation scenarios. Learners need to know how to assess different situations and perform first aid accordingly. In doing so, their clinical judgment and decision-making skills can improve.

Two students stated the following:

“Through the AR system, I can switch scenarios randomly, and the baby’s symptoms can help improve my judgment.”

“In the past, I simply learned the techniques required for first aid. However, AI made the baby’s reactions feel very real, which allowed me to learn how to assess an emergency and find the most optimal method.”

### 3.3. Stress-Free Learning Environment

With traditional teaching approaches, first aid skills are inevitably taught and demonstrated by instructors in person. However, this often causes learners to feel nervous, afraid, or unconfident, which can affect their performance. However, the AR system provides feedback to learners, guides them to take the correct first aid measures, and creates a self-directed and stress-free learning environment.

Two students stated the following:

“When instructors teach one-on-one, I always got nervous and felt afraid of making mistakes, but the learning environment provided by the AR system allows me to learn better without stress.”

“In the learning environment provided by the AR system, we can practice first aid without the presence of instructors, and so the learning pressure is greatly reduced.”

### 3.4. Improving the Efficiency of Distance- and Self-Learning

The AR application can be downloaded and used on any handheld mobile device, and learners can practice performing first aid repeatedly at any location and at any time. Two students stated the following:

“I can download the AR application on my cellphone and learn independently anytime and anywhere.”

“Due to the interactive features of the AR system, learning first aid becomes livelier and more interesting. It makes me want to practice first aid repeatedly at home.”

## 4. Discussion

This paper aimed to consolidate learners’ first aid skills in the removal of an infant’s airway obstruction; reduce delays in first aid treatment or prevent loss of life due to errors of judgment during an emergency; and enable learners to independently learn anytime and anywhere. Thus, this study adopts AR technology to develop a training model. With the training model, trainees can learn first aid more efficiently, correctly, and digitally conduct a quantitative assessment of their skills at the same time.

The AR-based assessment system for training in first aid for infant airway obstruction has two innovations. First, the system introduces AR technology; various emergency scenarios are designed with image targets created through Unity and Vuforia AR. Thus, the infant manikin can show expressions and symptoms, and make sounds on the mobile device, which helps learners with their clinical judgment and decision-making skills. Second, the built-in pressure sensor systems on the chest and back of the infant manikin are connected to the AR application through the wireless transmission system, which enables the assessment system to guide learners through proper first aid procedures and techniques. For instructors, this reduces the time required for assessment and the potential for human error. Previously, it was difficult to quantitatively assess first aid training, but now, with digital technology, the problem can be effectively solved.

However, some students expressed that because the AR application was a new learning mode, extra time was required to learn and adapt to it. Other students stated the greatest difference between the AR first aid skill learning module and traditional skill practice methods. The first aid skill learning module can provide a variety of first aid situations, and students’ judgment ability can be improved through repeated practice. The application of AR in nursing training can not only reduce the cost of training but also prevent patients from being exposed to a high-risk environment while allowing learners to receive training safely [6,16]. The AR-based assessment system provides various simulation scenarios, allowing learners to practice first aid and receive immediate feedback for an unlimited number of times, which in turn prevents human error and improves the quality of nursing care. In addition, under the impact of the epidemic, teachers and students cannot attend face-to-face classes, so both teaching and skill training are severely limited. However, with the AR system, students can start learning remotely by scanning the QR code with their tablet or cellphone. The purpose of flipped classrooms can therefore be fulfilled since students can practice first aid repeatedly and independently. AR is an effective learning tool for first aid training [9] since it can enhance students’ learning experiences and make learning more interesting [6]. This study has determined that AR provides a stress-free learning environment and that most of the students offered positive feedback. The students’ feedback shows that the AR skill education system improved their empathy, first aid skills, critical thinking skills, motivation, and confidence, and also reduced their anxiety about first aid training.

## 5. Conclusions

Currently, the typical manikins used in first aid training do not have a standardized and quantitative assessment system and cannot be used as a tool for digital education. Since there is an urgent need for nursing students, and even the public and medical staff, to receive efficient and correct first aid training, this study has adopted an AR-based assessment system for training in first aid for infant airway obstruction. The AR system can help learners improve both their first aid and critical thinking skills. Furthermore, the AR system allows learners to repeatedly practice first aid so that they can learn how to assess different emergencies and take appropriate measures; more importantly, learners are able to learn independently and repeatedly through this system without regard for time or space constraints. With the automatic assessment and AR-guided learning modes of the system, learners can learn first aid more efficiently and correctly. Instructors can also design various activities for first aid training with the system so that training can be lively and the digital learning and assessment system can become a mainstream practice in nursing education. In the next stage, we will continue to evaluate the quality of performance of students or nurses trained with the AR-based training compared to instructor-based training.

## 6. Patents

The name of the invention patent of the Taiwan (Republic of China): First Aid Training System Utilizing Augmented Reality [Application number:110114583]. The work has won a gold medal at the 33^RD^ International Invention, Innovation & Technology Exhibition 2022 (ITEX), Malaysia, 26–27 May 2022.

## Figures and Tables

**Figure 1 children-09-01622-f001:**
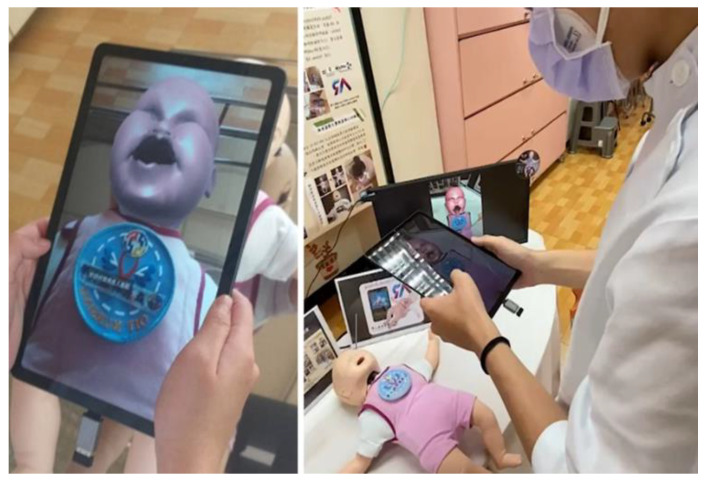
AR scanning system.

**Figure 2 children-09-01622-f002:**
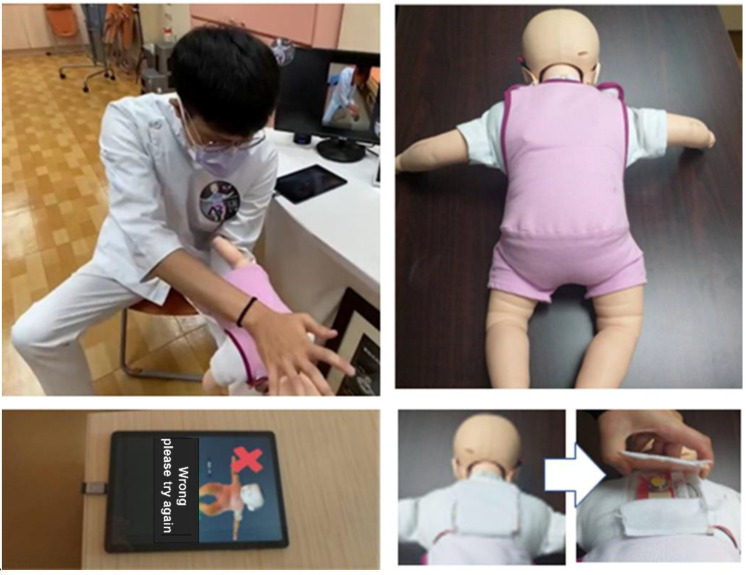
A pressure sensor for detecting back slaps.

**Figure 3 children-09-01622-f003:**
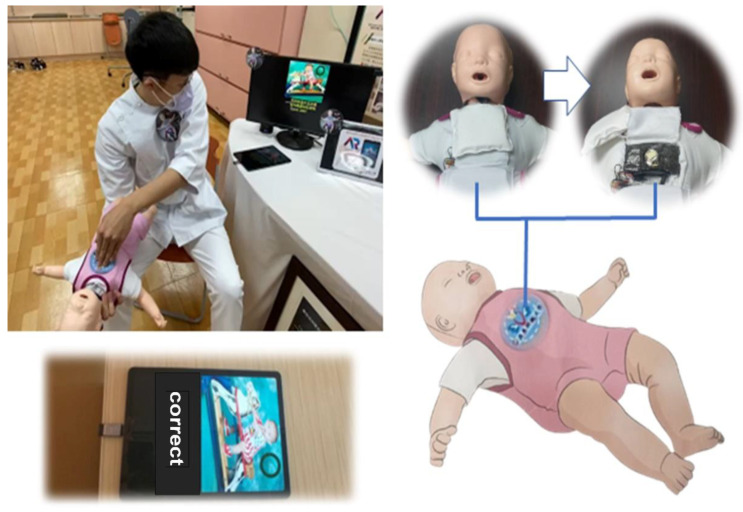
A pressure sensor for detecting chest thrusts.

**Figure 4 children-09-01622-f004:**
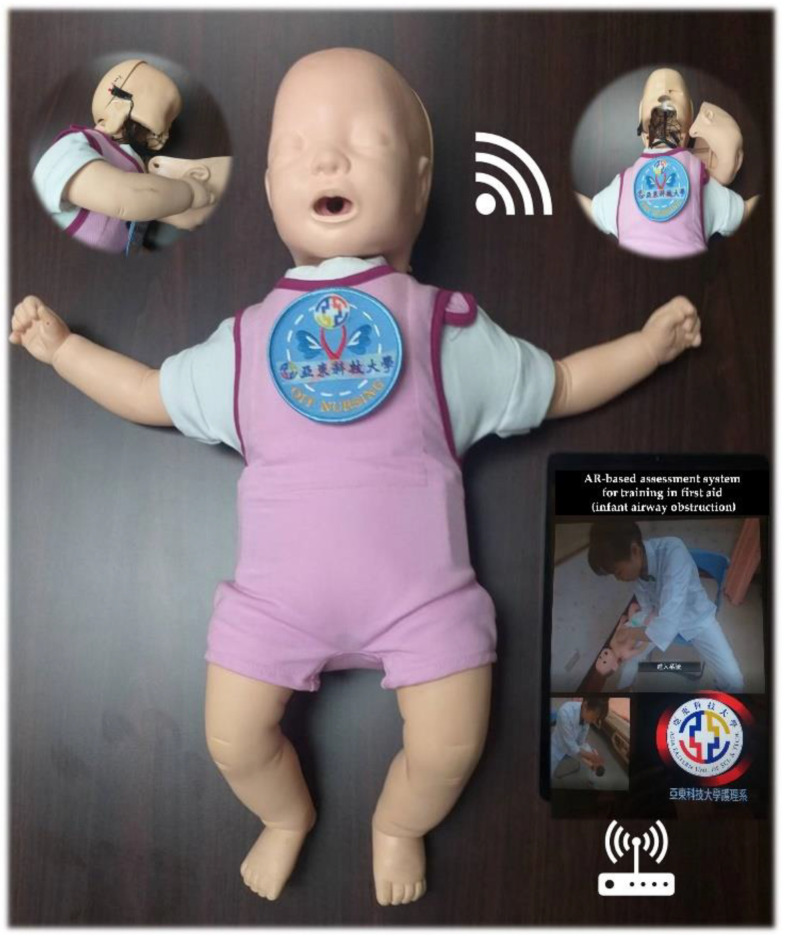
Wireless transmission system.

## Data Availability

The data that support the findings of this study are available on request from the first author, P.-J.C. The data are not publicly available due to their containing information that could compromise the privacy of research participants.

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
