# Peer review of "Development and Application of AR-Based Assessment System for Infant Airway Obstruction First Aid Training"

_children, 2022, doi:10.3390/children9111622_

Round 1

Reviewer 1 Report

Thank you for the opportunity to review the article "
 Development and Application of AR-Based Assessment System for Infant Airway Obstruction First Aid Training "The paper is interesting, but it must better structured, especially the Methods. The authors did not mention before the results section how the outcomes would be evaluated. The scales/methods of evaluation has to be clearly identified before. The authors quote the students in the results, but was never mentioned before if it was a qualitative study. Please, review the methods/structure 

Other comments
L31. Please modify the sentence « However, when too many students watch the demonstration too close to the instructors, the instructor may be interrupted while at the same time not all students can see clearly due to the crowded space. » The flow is a bit off.
L75. Remove the part of the sentence “With the training model, trainees can learn first aid more efficiently and correctly”. This is the introduction, and it has to be general
L76. Same the the sentence “For instructors, this reduces the time required for assessment and 77 the potential for human error ». Use it in discussion
Paragraph starting at line 71 is confusing and the flow must be improved. The authors stage 2 sets of different aims (L71) and L80.
Methods. Can the authors specify is the model was validated before / used before.
Results “This AR system was applied to the child nursing curriculum in April 2022. Among 156 the post-course reflections of 82 nursing students in the nursing department of a university in New Taipei City, Taiwan, most students provided positive feedback ». This must be stated first in the Methods section. The authors should add a paragraph regarding how /when it was evaluated

Author Response

Dear Reviewer,
A summary of major revisions made to the manuscript (using red text) is in the attachment. (Please see the attachment) Thank you very much for giving us this opportunity to revise the manuscript, and I greatly look forward to hearing from you soon.

Response to Reviewer 1 Comments

Point 1: Development and Application of AR-Based Assessment System for Infant Airway Obstruction First Aid Training "The paper is interesting, but it must better structured, especially the Methods. The authors did not mention before the results section how the outcomes would be evaluated. The scales/methods of evaluation have to be clearly identified before. The authors quote the students in the results but was never mentioned before if it was a qualitative study. Please, review the methods/structure

Response 1:

Thank you very much for your positive comments and careful reading of this manuscript. Based on your comments, we have made a comprehensive revision of the paper based on your comments. The methods section already elaborated on how the outcomes would be evaluated. (line number 86-151.)

2.1 Design and participants

       This study employs a qualitative research design and the research method of focus group interview, selecting a sample of 82 students from the Department of Nursing at a university in New Taipei City, Taiwan by purposive sampling. The participants' inclusion criteria are: a. Those who were willing to participate in this study and signed the informed consent form. b. Those who have normal cognitive function, no visual or hearing impairment, and can express themselves fluently through spoken language. c. The participants who were able to participate in the entire infant first aid training course and the focus group interviews.

2.2 Infant airway obstruction first aid basic skills

      First aid skills for infant airway obstruction were planned as part of the infant first aid training module of the pediatric nursing practical course. The instructor teaches students first aid skills for airway obstruction in infants based on the guidelines for providing emergency care for infants from the 2020 American Heart Association guidelines [12]. A child with a presumed airway obstruction that is still able to maintain some degree of ventilation should be allowed to clear the airway by coughing. If the child cannot cough, vocalize, or breathe, emergent steps are necessary to clear the airway. For infants under one year of age, alternating sequences of five back blows and five chest thrusts are performed until the object clears or the infant becomes unresponsive. Abdominal thrusts should not be performed on infants as their livers are more prone to injury.

2.3 Data collection procedure

      The course instructor first demonstrated the first aid procedure and the key points of back blows and chest thrusts, emphasizing that the infant's head and neck must be sufficiently fixated with one hand during the first aid procedure to avoid intracranial hemorrhage due to shaking. Students were allowed to practice with traditional infant choking manikins. Afterwards, students would experience the “AR Infant Airway Obstruction First Aid Assessment System” and repeat the exercise using the interactive model developed in this study. At the end of the practical sessions, the research method of focus group was used to understand individual attitudes, perspectives, or beliefs about a topic, thus allowing researchers to gain insight into interesting situations and to clarify the meaning behind certain behaviors [13]. The focus group interviews were held from the 12th to the 14th of April in 2022 and consisting of six 50-minute sessions with approximately 13 to 14 participants. The interviews took place in a quiet and comfortable discussion room. In order to protect the rights of the students, the informed consent form states that students’ participation in AR learning and interviews is not related to their academic performance.

      During the interviews, the students were asked about their perceptions of utilizing the AR system to learn first aid skills and reflect on its impact on learning. The researchers confirmed the content and the perceptions provided by the participants and ensured that the participants had sufficient time to fully answer each question, thus ensuring the credibility of the study. In addition to discussions on the content, the interview records also included the participants’ nonverbal behavior or emotions. The interview outline was first developed by the researchers through a literature review; subsequently, the interview questions were finalized based on their individual teaching experience and the consensus of the research team. The course instructor was not the moderator of the focus group interview sessions. The moderator of the interview sessions was another member of the research team who was not the instructor of the first aid course.

The guiding questions of the interview are as follows:

  1. Please share your feelings about the learning experience of first aid skills through the AR system.
  2. What is the difference between practicing first aid skills and traditional skills in an AR learning environment?
  3. How did the AR system help or benefit you in learning first aid skills?
  4. Did you encounter any difficulties in learning first aid skills with the AR system?
  5. In the face of the trend of integrating digital technology into the curriculum, do you have any suggestions for the usage of AR in teaching?

2.4 Data analysis

       In this study, content analyze was used to analyses the data, and the steps were as follows: first, the audio recordings of the focus sessions were translated verbatim without any changes or additions; afterwards, the researchers read the data repeatedly to search for the categories presented in the data and categorize the data. Lastly, during the coding process, the codes of the same nature were connected to form categories, and the links between the categories were further unified to form the main theme [14,15]. Two experts trained in qualitative research were invited to conduct a peer examination, and five interviewees were asked to conduct a member check to confirm whether the data analysis results represented their real learning experiences. The NVivo 10 software program was used to code the data according to the themes and extract key data from each theme.

(line number 86-151.)

Point 2: L31. Please modify the sentence « However, when too many students watch the demonstration too close to the instructors, the instructor may be interrupted while at the same time not all students can see clearly due to the crowded space. » The flow is a bit off.

Response 2:

Based on your comments, we have revised the sentence as follows:

Due to the crowded space in the classroom, not all students can see the demonstration by the instructors clearly, and the demonstration also may be interrupted by the student. (line number 31-33.)

Due to the crowded space in the classroom, not all students can see the demonstration by the instructors clearly, and the demonstration may also be interrupted by students.

(line number 31-33.)

Point 3: L75. Remove the part of the sentence “With the training model, trainees can learn first aid more efficiently and correctly”. This is the introduction, and it has to be general.

Response 3:

We already moved the sentence " With the training model, trainees can learn first aid more efficiently and correctly." to the discussion section. (line number 278-283.)

      To consolidate learners' first aid skills in the removal of an infant’s airway obstruc-tion; reduce delays in first aid treatment or prevent loss of life due to errors of judgment during an emergency; and enable learners to independently learn anytime and anywhere. Thus, this study adopts AR technology to develop a training model. With the training model, trainees can learn first aid more efficiently, correctly and digitally conduct a quan-titative assessment of their skills at the same time.

(line number 278-283.)

Point 4:  L76. Same the the sentence “For instructors, this reduces the time required for assessment and 77 the potential for human error ». Use it in discussion

Paragraph starting at line 71 is confusing and the flow must be improved.

Response 4:

1.We already moved the sentence "For instructors, this reduces the time required for assessment and the potential for human error." to the discussion section. (line number 292-294.)

For instructors, this reduces the time required for assessment and the potential for human error. Previously, it was difficult to quantitatively assess first aid training, but now, with digital technology, the problem can be effectively solved. (line number 292-294.)

2.We already rewrote lines 73 to 84 to be a new paragraph.

      Because of the unique anatomy and physiology of infants and toddler children, airway obstruction by foreign bodies is one of the leading causes of accidental death [10]. Besides complete foreign body airway obstruction that is immediately life-threatening, partial airway obstructions can impede gas exchange and lead to dyspnea and pneumonia [11]. Thus, early recognition, emergency care, and management are essential. The objectives of this study are to develop an AR-based assessment system for training in first aid for infant airway obstruction and an interactive manikin. This system, combined with the manikin, can guide learners through first aid training step-by-step and offer immediate feedback to ensure that their procedures are completely correct, and then record and quantify their performance and provide immediate evaluation. Furthermore, this study also attempted to understand students’ learning experiences toward self-learning using the AR skill education system.

(line number 73-84.)

 Point 5:. authors stage 2 sets of different aims (L71) and L80.

Response 5:

Thank you very much for your careful reading of this manuscript and comments.

We already rewrote lines 73 to 84 to be a new paragraph.

Point 6: Methods. Can the authors specify is the model was validated before / used before. Results “This AR system was applied to the child nursing curriculum in April 2022. Among 156 the post-course reflections of 82 nursing students in the nursing department of a university in New Taipei City, Taiwan, most students provided positive feedback ». This must be stated first in the Methods section. The authors should add a paragraph regarding how /when it was evaluated

Response 6:

Thank you for your suggestions and detailed comments to improve the quality of this paper.

The methods section already elaborated on how the outcomes would be evaluated.

(line number 86-151.)

Reviewer 2 Report

The presented manuscript describes the invention of a new augmented-reality based resuscitation training in a scenario with airway obstruction in infants with an interactive manikin providing immediate feedback on the taken measures. This training aims at improvement of skills in the removal of the airway obstruction, fast first-aid skills and recognition of emergencies. The use of this model enables training of students, nurses or other medical personal with an immediate feedback from pressure sensors in the manikin. Feedback from participants after first application were very positive.

The manuscript is easy to read and to understand as it is purely descriptive without further scientific assessment. It presents a new approach to integrate new technologies as AR in the training of medical personal. By that, scenarios are much more realistic and the learning of different scenarios with audio feedback of coughing and choking, progressive cyanosis help the students to have a faster recognition of upcoming emergencies.

I wonder, whether the presented manikin has an accelerator-sensor in the head, as with back-thrusts, the head can tilt back and forth with very much force possibly resulting in contusion cerebri and even hemorrhages. This would be a very important feedback for the providers of first aid measures. If this is not implemented the presence of an instructor is clearly required to ensure, that the head is sufficiently fixated with one hand during back-thrusts.

In my opinion, this AR based training and interactive manikin has a high potential to improve resuscitation skills therefore a publication of the manuscript can be encouraged to spread the knowledge of these innovative teaching possibilities. I do not think that it is possible to learn the process completely without an instructor (e.g. handling an infant in this situation with turning from back to belly and vice versa). The authors of the manuscript should discuss how the basic skills are taught.

In a next step, the authors should to evaluate the quality of performance of students or nurses trained with the AR-based training compared to an instructor based training which could be very easily evaluated with their manikin. 

One minor point: the authors often use the term “golden hour” which is used very often in critical situations. However, with the given manikin the training is a basic-life support scenario, not an advanced life support scenario. Airway obstruction in infants is a highly acute emergency within less than 10 minutes. The use of the term “golden hour” might imply that we have a lot of time in this setting. I would suggest to critically question the use of this term.

Suggestion: perhaps one short sample scenario (audio/video) could be uploaded to a supplemental file to give insight in the possibilities

Author Response

Dear Reviewer,
A summary of major revisions made to the manuscript (using red text) is in the attachment. (Please see the attachment) Thank you very much for giving us this opportunity to revise the manuscript, and I greatly look forward to hearing from you soon.

Response to Reviewer Comments

Point 1: The presented manuscript describes the invention of a new augmented-reality based resuscitation training in a scenario with airway obstruction in infants with an interactive manikin providing immediate feedback on the taken measures. This training aims at improvement of skills in the removal of the airway obstruction, fast first-aid skills and recognition of emergencies. The use of this model enables training of students, nurses or other medical personal with an immediate feedback from pressure sensors in the manikin. Feedback from participants after first application were very positive.

The manuscript is easy to read and to understand as it is purely descriptive without further scientific assessment. It presents a new approach to integrate new technologies as AR in the training of medical personal. By that, scenarios are much more realistic and the learning of different scenarios with audio feedback of coughing and choking, progressive cyanosis help the students to have a faster recognition of upcoming emergencies.

Response 1:

Dear reviewer, thank you so much for your suggestions and detailed comments to improve the quality of this paper. I am deeply grateful for the thoughtful reviews and your kind help. We have made a comprehensive revision of the paper based on your comments. Thank you very much for giving us this opportunity to revise the manuscript, and I greatly look forward this paper could be suitable for published to the community.

Point 2: I wonder, whether the presented manikin has an accelerator-sensor in the head, as with back-thrusts, the head can tilt back and forth with very much force possibly resulting in contusion cerebri and even hemorrhages. This would be a very important feedback for the providers of first aid measures. If this is not implemented the presence of an instructor is clearly required to ensure, that the head is sufficiently fixated with one hand during back-thrusts.

Response 2:

Thank you very much for your positive comments, we have added in our manuscript that the instructor is clearly required to ensure that the head is sufficiently fixated with one hand during back-thrusts. (line number 107-110.)

The course instructor first demonstrated the first aid procedure and the key points of back blows and chest thrusts, emphasizing that the infant's head and neck must be sufficiently fixated with one hand during the first aid procedure to avoid intracranial hemorrhage due to shaking. (line number 107-110.)

Point 3: In my opinion, this AR based training and interactive manikin has a high potential to improve resuscitation skills therefore a publication of the manuscript can be encouraged to spread the knowledge of these innovative teaching possibilities. I do not think that it is possible to learn the process completely without an instructor (e.g. handling an infant in this situation with turning from back to belly and vice versa). The authors of the manuscript should discuss how the basic skills are taught.

Response 3:

Thank you for your very helpful suggestions. We've included content on how basic skills are taught in our manuscript.

We already add a paragraph regarding how the basic skills are taught as follow.

(line number 95-105.)

2.2 Infant airway obstruction first aid basic skills

       First aid skills for infant airway obstruction were planned as part of the infant first aid training module of the pediatric nursing practical course. The instructor teaches students first aid skills for airway obstruction in infants based on the guidelines for providing emergency care for infants from the 2020 American Heart Association guidelines [12]. A child with a presumed airway obstruction that is still able to maintain some degree of ventilation should be allowed to clear the airway by coughing. If the child cannot cough, vocalize, or breathe, emergent steps are necessary to clear the airway. For infants under one year of age, alternating sequences of five back blows and five chest thrusts are performed until the object clears or the infant becomes unresponsive. Abdominal thrusts should not be performed on infants as their livers are more prone to injury. (line number 95-105.)

Point 4: In a next step, the authors should to evaluate the quality of performance of students or nurses trained with the AR-based training compared to an instructor-based training which could be very easily evaluated with their manikin.

Response 4:

  1. We appreciate your professional advice and assistance and we have added a paragraph to the discussion section. (line number 295-300.)

However, some students expressed that because the AR application was a new learning mode. Therefore, extra time was required to learn and adapt to it. Other students stated that the greatest difference between the AR first aid skill learning module and traditional skill practice methods. The first aid skill learning module can provide a variety of first aid situations, and students' judgment ability can be improved through repeated practice.  (line number 295-300.)

  1. We have also added a paragraph to the conclusions section in our manuscript as follows: (line number 330-332.)

In the next stage, we will continue to evaluate the quality of performance of students or nurses trained with the AR-based training compared to instructor-based training. (line number 330-332.)

Point 5: One minor point: the authors often use the term “golden hour” which is used very often in critical situations. However, with the given manikin the training is a basic-life support scenario, not an advanced life support scenario. Airway obstruction in infants is a highly acute emergency within less than 10 minutes. The use of the term “golden hour” might imply that we have a lot of time in this setting. I would suggest to critically question the use of this term.

Response 5:

Thank you for your guidance and reminder. We have revised the "golden hour" to be "urgent time." (line number 27, 155 and 216)

Point 6: Suggestion: perhaps one short sample scenario (audio/video) could be uploaded to a supplemental file to give insight in the possibilities

Response 6:

Thank you for your very creative comments. We already uploaded a short sample video to a supplemental file and provided a video link as follows, which could give our reader more insight.

https://www.youtube.com/watch?v=pzZ_dQaEFi4

(line number 220-221 in 2.5.5 System operation process section; line number 338-339 in Supplementary Materials)

Round 2

Reviewer 1 Report

Thank you for answering my comments.

I have one minor comment 

"L35.  I would remove the last part of the sentence “the demonstration may also be interrupted by students”. It sounds like it is not a good thing to have interaction during classroom."